

# Partitioning of canopy and soil CO₂ fluxes in a pine forests at the dry timberline

**Rafat Qubaja[a], Feyodor Tatarinov[a], Eyal Rotenberg[a], and Dan Yakir[a]***

[a] Department of Earth and Planetary Sciences, Weizmann Institute of Science, Rehovot 76100, Israel

*Correspondence:* Dan Yakir; email: dan.yakir@weizmann.ac.il

**Abstract**

Partitioning carbon fluxes is key to understanding the process underlying ecosystem response to change. This study used soil and canopy fluxes with stable isotopes ($^{13}$C) and radiocarbon ($^{14}$C) measurements of a 50-year-old dry (i.e., 287 mm of annual precipitation) pine forest to partition the ecosystem's CO₂ flux into gross primary productivity (GPP) and ecosystem respiration (Re) and soil respiration flux into autotrophic (Rsa), heterotrophic (Rh), and inorganic (Ri) components. On an annual scale, GPP and Re were 655 and 488 g C m$^{-2}$, respectively, with a net primary productivity (NPP) of 276 g C m$^{-2}$ and carbon-use efficiency (CUE=NPP/GPP) of 0.42. Soil respiration (Rs) made up 60 % of the total ecosystem respiration and was comprised of 24 ± 4 %, 23 ± 4 %, and 13 ± 1 % Rsa, Rh, and Ri, respectively. The contribution of root and microbial respiration to Re increased during high productivity periods, and inorganic sources were more significant components when soil water content was low. Compared to the mean values for 2001-2006 at the same site; Grünzweig et al., 2009), annual Rs decreased by 27 % to the mean 2016 rates of 0.8 ± 0.1 µmol m$^{-2}$ s$^{-1}$). This was associated with decrease in the respiration $Q_{10}$ values across the same observation by 36 % and 9 % in the wet and dry periods, respectively. Low rates of soil carbon loss combined with relatively high below ground carbon allocation (i.e., 40 % of canopy CO₂ uptake) help explain the high soil organic carbon accumulation and the relatively high ecosystem CUE of the dry forest. This was indicative of the higher resilience of the pine forest to climate change and the significant potential for carbon sequestration in these regions.

**Keywords**: Carbon balance, Soil respiration, Autotrophic, Heterotrophic, Inorganic flux, Temperature response, Semi-arid ecosystem, Pine forest, Canopy cover, and Soil chamber.





## 1. Introduction

On a global scale, soil stores over 1,500 Pg of carbon, which is more than the atmosphere and terrestrial plant biomasses combined (Köchy et al., 2015; Le Quéré et al., 2018; Scharlemann et al., 2014). Soil respiration (Rs) from terrestrial ecosystems and the atmosphere constitutes a large part of the terrestrial carbon cycle (Le Quéré et al., 2018), releasing 68–98 Pg C into the atmosphere as $CO_2$ annually (Adachi et al., 2017; Hashimoto et al., 2015; Zhao et al., 2017). This is far more than fossil fuel emissions by an order of magnitude (Ballantyne et al., 2015).

The annual net storage of carbon in land biospheres, known as net ecosystem production (NEP), is the balance between carbon uptake during gross primary productivity (GPP), carbon loss during growth and maintenance respiration by plants (i.e., autotrophic respiration [Ra]), and decomposition of litter and soil organic matter (i.e., heterotrophic respiration [Rh]; Bonan, 2008; Chapin et al., 2006; Schulze, 2006). The difference between GPP and Ra is expressed as net primary production (NPP) and is the net carbon uptake of plants used for new biomass production. Measurements from a range of ecosystems have shown that total plant respiration can be as much as 50 % of GPP (Ryan, 1991), and together with Rh comprises total ecosystem respiration (Re, Re = Ra + Rh). The partitioning of ecosystem carbon fluxes can therefore be summarized as:

$$GPP = NPP + Ra = NEP + Rh + Ra. \tag{1}$$

Earlier campaign-based measurements carried out by Maseyk et al. (2008a) and Grünzweig et al. (2009) in the semi-arid *Pinus halepensis* (Aleppo pine) Yatir forest indicated that GPP at this site was lower than that among temperate coniferous forests (1000–1900 g C $m^{-2}$ $y^{-1}$) but within the range estimated for Mediterranean evergreen and boreal coniferous forests (Falge et al., 2002) with high carbon-use efficiency of 0.4 (CUE = NPP/GPP; DeLucia et al., 2007). The total flux of $CO_2$ released from the ecosystem (Re) is partitioned into aboveground autotrophic respiration (i.e., through foliage and sapwood [Rf]) and soil $CO_2$ flux (Rs). Rs, in turn, is a combination of three principal components and can be further partitioned into the components originating from roots or rhizospheres and mycorrhizas (i.e., belowground autotrophic [Rsa]), from carbon respired during the decomposition of dead organic matter by soil microorganisms and macrofaunal (heterotrophic respiration [Rh]; Bahn et al., 2010; Kuzyakov, 2006; Ryan and Law, 2005), and from pedogenic or anthropogenic acidification of soils containing $CaCO_3$ (Ri; Heinemeyer et al., 2007; Kuzyakov, 2006), which is expressed as





$Re = Rs + Rf = [Rsa + Rh + Ri] + Rf$                    (2)

Previously published results show that Rsa contribution to Rs ranges from 24–65 % in forest soils in
different biomes and ecosystems (Andersen et al., 2005; Binkley et al., 2006; Chen et al., 2010; Frey et al.,
2006; Johnsen et al., 2007; Hogberg et al., 2009; Olsson et al., 2005; Subke et al., 2011). However, most of
these experiments were performed in boreal, temperate, or subtropical forests, and there is a general lack
of information on water-limited environments, such as dry Mediterranean ecosystems.
The link between soil $CO_2$ efflux and respiration is complex. For example, a considerable fraction of the
respired $CO_2$ can be dissolved in soil water, transported in hydrological systems, or take part in reactions
of carbonate systems. In a calcareous soil with a pH of ~8, most system carbon is bicarbonate ($HCO_3^-$),
while in calcareous soils, $CO_2$ can be consumed during calcium–carbonate dissolution reactions or released
during reverse reactions as carbonate precipitation (Benavente et al., 2010; Cuezva et al., 2011; Kowalski
et al., 2008). Processes within root $CO_2$ can be dissolved in xylem water and carried upward through the
transpiration stream (Aubrey and Teskey, 2009; Bloemen et al., 2014).
Soil carbon in semi-arid regions may be strongly influenced by soil inorganic carbon (SIC; Schlesinger,
1982). Its dissolution (which creates a sink for atmospheric $CO_2$) deposition (i.e., source), or recycling can
contribute to soil $CO_2$ fluxes. On annual to decadal time scales, this contribution is assumed to be marginal
at 3–4 g C $m^{-2}$ $y^{-1}$, compared to 60 ± 6 g C $m^{-2}$ $y^{-1}$ for tundra and up to 1260 ± 57 g C $m^{-2}$ $y^{-1}$ for tropical
moist forests (Raich and Schlesinger, 1992). This small abiotic flux can be accepted as an insignificant
source compared to biotic $CO_2$ sources. However, uncertainties concerning the significance of the abiotic
process for C budgets in dry ecosystems with calcareous soils exist.
Rates of Rs, have been altered due to global climatic change, particularly through changes in soil
temperature (Ts) and soil moisture (SWC; Bond-Lamberty and Thomson, 2010; Buchmann, 2000;
Carvalhais et al., 2014; Davidson et al., 1998; Zhou et al., 2009), which account for 65–92 % of the
variability of Rs in a mixed deciduous forest (Peterjohn et al., 1994). $CO_2$ efflux generally increases with
increasing soil temperatures (Frank et al., 2002) and can produce positive feedback for climate warming
(Conant et al., 1998), converting a biosphere from a net carbon sink to a carbon source (IPCC-AR5 2014).
A range of empirical models have been developed to relate Rs rate and temperature (Balogh et al., 2011;
Lellei-Kovács et al., 2011), and the most widely used model relies on the $Q_{10}$ approach (Bond-Lamberty
and Thomson, 2010), which quantifies the sensitivity of Rs to temperature and integrates physical
processes, such as rate of $O_2$ diffusion into and $CO_2$ diffusion out of soils and the intrinsic temperature





dependency of enzymatic processes (Davidson and Janssens, 2006). Soil moisture (SWC) may be of greater
importance than temperature in influencing Rs in water-limited ecosystems (Cable et al., 2011; Grünzweig
et al., 2009; Shen et al., 2008; Saleska et al., 1999). In general, the Rs rate increases with the increase of
SWC at low levels but decreases at high levels of SWC (Deng et al., 2012; Hui and Luo, 2004; Jiang et al.,
2013). Several studies have connected the sensitivity of carbon fluxes in semi-arid Mediterranean
ecosystems to the irregular seasonal and interannual distribution of rain events (Poulter et al., 2014; Ross
et al., 2012). While Rs is generally constrained by low SWC during summer months, abrupt and large soil
$CO_2$ pulses have been observed after rewetting dry soil (Matteucci et al., 2014).
Partitioning ecosystem $CO_2$ fluxes using stable isotopes has been proposed as a powerful partitioning
approach (Ogee et al., 2004; Yakir and Sternberg, 2000). Earlier studies in the semi-arid Yatir forest
indicated that using $^{13}C$ creates net ecosystem carbon losses in summer, which are driven by soil emission
(Maseyk et al., 2008a), and that different sources of soil carbon can be identified (Grünzweig et al., 2007).
Using both $^{13}C$ and $CO_2/O_2$ ratios also showed that abiotic processes, such as $CO_2$ storage, transport, and
interactions with sediments, can influence Rs measurements at this site (Angert et al., 2015; Carmi et al.,
2013). The current research provides follow-up measurements to the 2001-2006 study at the same site to
identify long-term temporal changes in the soil–atmosphere $CO_2$ fluxes in this environment. It also extends
the earlier studies by using continuous soil and ecosystem flux measurements and auxiliary analyses to
fully and quantitatively partition soil $CO_2$ to better understand the high carbon sequestration potential in
this semi-arid forest planation, and.
**2. Materials and methods**
**2.1. Site description**
The Yatir forest (3˚20' N, 35˚03' E) is located in the transition zone between sub-humid and arid
Mediterranean climates (Fig. S1) on the edge of the Hebron mountain ridge at a mean altitude of 650 m.
The ecosystem is a semi-arid pine afforestation established in the 1960s and covering approximately 18
km². The average air temperatures for January and July are 10 ºC and 25.8 ºC, respectively. Mean annual
potential evapotranspiration (ET) is 1,600 mm, and mean annual precipitation is 285 mm. Only winter
(December to March) precipitation occurs in this region, creating a distinctive wet season, while summer
(June to October) is an extended dry season. There are short transition periods between seasons, with a
wetting season (i.e., autumn) and a drying season (i.e., spring). The forest is dominated by Aleppo pine
(*Pinus halepensis*), with smaller proportions of other pine species and cypress and little understory





vegetation. Tree density in 2007 was 300 trees ha$^{-1}$; mean tree height was 10 m; and native background
vegetation was sparse shrubland with a total vegetation height of 0.30–0.50 m (Grünzweig et al., 2003).
The soil at the research site is shallow (20–40 cm) Aeolian-origin loess with a clay-loam texture (31% sand,
41 % silt, and 28% clay; density: $1.65 \pm 0.14$ g cm$^{-3}$) overlying chalk and limestone bedrock. Deeper soils
(up to 1.5 m) are sporadically located at topographic hollows. While the natural rocky hill slopes in the
region are known to create flash floods, the forested plantation reduces runoff dramatically to less than 5%
of annual rainfall (Shachnovich et al., 2008). Groundwater is deep (> 300 m), reducing the possibility of
groundwater recharge due to negative hydraulic conductivity or of water uptake by trees from the
groundwater.
**2.2. Flux and meteorological measurements**
An instrumented eddy covariance tower was erected in the geographical center of Yatir forest, following
the EUROFLUX methodology (Aubinet et al., 2000). The system uses a three-dimensional (3D) sonic
anemometer (Omnidirectional R3, Gill Instruments) and a closed path LI-COR 7000 $CO_2/H_2O$ gas analyzer
(LI-COR Inc., Nebraska, USA) to measure the evapotranspiration flux (ET) and net $CO_2$ flux (NEE). EC
flux measurements were used to estimate the annual scale of NEP by integrating half-hour NEE values. The
15-year NEE record was obtained after U* night-time correction, gap filling, and quality control as
described in Tatarinov et al. (2016). A site-specific algorithm was used for flux partitioning into Re and
GPP. Daytime ecosystem respiration (Re-d, in μmol m$^{-2}$ s$^{-1}$) was estimated based on measured night-time
(i.e., when the global radiation was < 5 W m$^{-2}$) values (Re-n), averaged for the first three half-hours of each
night. The daytime respiration for each half-hour was calculated according to Eq. 3 (Maseyk et al., 2008a;
Tatarinov et al., 2016),
$$R_{e-d} = R_{e-n}(\alpha_1\beta_s^{dT_s} + \alpha_2\beta_w^{dT_a} + \alpha_3\beta_f^{dT_a}) \tag{3}$$
where $\beta_s$, $\beta_w$, and $\beta_f$ are coefficients that correspond to soil, wood, and foliage, respectively; $dT_s$ and $dT_a$
are soil and air temperature deviations from the values at the beginning of the night; and $\alpha_1$, $\alpha_2$, and $\alpha_3$ are
partitioning coefficients fixed at 0.5, 0.1, and 0.4, respectively. The $\beta_s$, $\beta_w$, and $\beta_f$ coefficients were
calculated as follows: $\beta_s$ = 2.45 for wet soil (i.e., soil water content in the upper 30 cm above 20% vol); $\beta_s$
= 1.18 for dry soil (i.e., based on $Q_{10}$ from the Grünzweig et al. [2009] study at the same site); $\beta_f$ = 3.15–
0.036Ta; and $\beta_w$ = 1.34 + 0.46 exp (-0.5((DoY-162)/66.1)$^2$), where DoY is the day of the hydrological year





starting from October first. Finally, GPP was calculated as GPP = NEE-Re. Negative values of the NEE
and GPP indicated that the ecosystem was a $CO_2$ sink.
Half-hour auxiliary measurements used in this study included photosynthetic activity radiation (PAR mol
$m^{-2} s^{-1}$), vapor pressure deficit (VPD, kPa), wind speed (m $s^{-1}$), and relative humidity (RH, %)— additional
measurements are described elsewhere (Tatarinov et al., 2016). Air temperature (Ta, °C), relative humidity
(RH, %), and soil temperature (Ts, °C) were also measured and calculated using soil chambers 20 cm above
the soil surface and at a 5 cm depth. These were located at 21 points and measured every half hour using
soil chamber system (LI8150-203, LI-COR Lincoln, NE). Volumetric soil water content ($SWC_{0-10}$) was
measured in the upper 10 cm of the soil very half hour near the chambers using the ThetaProbe model
ML2x (Delta-T Devices Ltd., Cambridge, UK) calibrated to the soil composition based on the
manufacturer's equations.
**2.3. Soil $CO_2$ flux**
Soil $CO_2$ flux (Rs) was measured using automated non-steady-state systems, 20 cm diameter opaque
chambers, and a multiplexer to allow for simultaneous control of several chambers (LI -8150, -8100-101, -
8100-104; LI-COR, Lincoln, NE). Precision of $CO_2$ measurements in chamber air was ±1.5 % of the
measurements range (0–20,000 ppm). The chamber closed onto preinstalled PVC collars 20 cm in diameter,
inserted 5 cm into the soil and 6 cm above the surface, allowing for short measurement times (i.e., 2 min).
When measurements were not being taken, chambers were positioned away from collars. Data were
collected using a system in which air from the chamber was circulated (2.5 l $min^{-1}$) through an infrared gas
analyzer (IRGA) to record $CO_2$ (µmol $CO_2$/mol air) and $H_2O$ (mmol $H_2O$/mol air) concentrations in the
system logger (1 $s^{-1}$).
The rates of soil $CO_2$ flux or Rs were calculated from chamber data using a linear fit of change in the water-
corrected $CO_2$ mole fraction and Eq. 4 (LiCor Manual, 2015) as follows:

$$R_s = \frac{dC}{dt} \cdot \frac{v\,P}{s\,T_a R} \quad .$$  (4)

Here, Rs is the soil $CO_2$ flux (µmol $CO_2$ $m^{-2}s^{-1}$), dC/dt is the rate of change in the water-corrected $CO_2$ mole
fraction (µmol $CO_2$ $mol^{-1}$ air $s^{-1}$), v is the system volume ($m^3$), P is the chamber pressure (Pa), s is the soil
surface area within the collar ($m^2$), Ta is the chamber air temperature (K), and R is the gas constant (J $mol^-$




[1] $K^{-1}$). A measurement period of 2 minutes was used based on preliminary tests to obtain the most linear
increase of $CO_2$ in the chambers with the highest $R^2$.
Soil $CO_2$ fluxes in the experimental plot were measured between November 2015 and October 2016 with
three chambers using 21 collars arranged in seven groups (sites) on a half-hour basis (48 daily records).
The three chambers were rotated between the seven sites every 1–2 weeks to cover all sites and to assess
spatial and temporal variations.
Upscaling of the collar measurements to plot-scale soil $CO_2$ flux was carried out by grouping collars based
on the distance from trees (Dt) (i.e., under trees [< 1 m from nearest tree; UT], in gaps between trees [1–
2.3 m; BT], and open areas [> 2.3 m; OA]). One chamber was measured at each microsite group to estimate
the fractional areas (Ø) of the three locations based on mapping the sites according to the distances noted
above, which was previously done (Raz-Yaseef et al., 2010).

$$Rs = Rs_{OA} * Ø_{OA} + Rs_{BT} * Ø_{BT} + Rs_{UT} * Ø_{UT} \tag{5}$$
$$Ø_{OA} + Ø_{BT} + Ø_{UT} = 1 \tag{6}$$

The annual scale of Rs was derived from the up-scaled chamber based on daily records (48 half hourly
values) of spatial up-scaled Rs. Gap filling of missing data due to technical problems (i.e., 27 % of the data
across the study period of 2015–2016) was based on the average diurnal cycle of each month, and such data
were averaged to obtain the estimate of the annual scale of Rs.

Estimating the temperature sensitivity of Rs ($Q_{10}$) was done as described by Davidson and Janssens (2006)
using a first-order exponential equation (see also Xu et al., 2015),

$$Rs = ae^{bTs}, \tag{7}$$

where Rs represents the half-hour spatial up-scaled time series of soil respiration flux ($\mu$mol m$^{-2}$ s$^{-1}$), Ts
(°C) is soil temperature at a 5 cm depth (up-scaled spatially and temporally using the same method as Rs),
and a and b are fitted parameters. The b values were used to calculate the $Q_{10}$ value according to the
following equation:
$$Q_{10} = e^{10b}. \tag{8}$$





### 2.4. Soil CO$_2$ flux partitioning

Determination of different sources of soil CO$_2$ efflux was based on linear mixing models (Lin et al., 1999)
to estimate proportions for three main sources (autotrophic, heterotrophic, and abiotic), using isotopic
analysis of soil CO$_2$ profiles and soil incubation data from eight campaigns (January to September) during
2016 according to Equations 9–11. Partitioning of the monthly Rs values into components was done using
a 3-endmember triangular model for interpreting the $\delta^{13}$C and $\Delta^{14}$C values of CO$_2$ flux; the 3-endmember
triangular corners are the autotrophic (Rsa), heterotrophic (Rh), and abiotic (Ri) sources of Rs. The $\delta^{13}$C
and $\Delta^{14}$C isotope signatures of monthly Rs are located inside the triangle (Fig. S2).

$$\delta^{13}C_{Rs} = f_{sa} * \delta^{13}C_{sa} + f_h * \delta^{13}C_h + f_i * \delta^{13}C_i \qquad (9)$$
$$\Delta^{14}C_{Rs} = f_{sa} * \Delta^{14}C_{sa} + f_h * \Delta^{14}C_h + f_i * \Delta^{14}C_i \qquad (10)$$
$$1 = f_{sa} + f_h + f_i \qquad (11)$$

Here, f indicates the fraction of total soil flux (e.g., $f_h = R_h/Rs$), while subscripts sa, h, and i indicate
autotrophic, heterotrophic, and inorganic components, respectively. The three equation systems were used
to solve the three unknown f fractions of the total soil flux based on empirical estimates of the isotopic end-
members. Additionally, $\delta^{13}$ and $\Delta^{14}$ are the stable and radioactive carbon isotopic ratios, where $\delta^{13}C =$
$[([^{13}C/^{12}C]_{sample}/[^{13}C/^{12}C]_{refrence}) - 1] * 1000‰$, referencing the Vienna international standard (VPDB).
Radiocarbon data are expressed as $\Delta^{14}$C in parts per thousand or per mil (‰), which is the deviation of a
sample $^{14}$C/$^{12}$C ratio relative to the OxI standard in 1950 (see Taylor et al., 2015), that is $\Delta^{14}C =$
$[([^{14}C/^{12}C]_{sample}/(0.95 * [^{14}C/^{12}C]_{refrence} * \exp[(y - 1950)/8267])) - 1] * 1000‰$, where y is the year of
sample measurement.
The $\delta^{13}C_{Rs}$ was estimated monthly using the Keeling plot approach (Figs. S3 and 4; Pataki et al., 2003;
Taneva and Gonzalez-Meler, 2011). Soil air was sampled using closed-end stainless steel tubes (6 mm
diameter) perforated near the tube bottom at four depths (30, 60, 90, and 120 cm). Samples of soil air were
collected in pre-evacuated 150 mL glass flasks with high-vacuum valves, with the dead volume in the
tubing and flask necks purged with soil air using a plastic syringe equipped with a three-way valve. The
$\delta^{13}C_{sa}$ end-member was estimated based on incubations during the sampling periods of excised roots
following Carbone et al. (2008). Fine roots (< 2 mm diameter) were collected, rinsed with deionized water,
and incubated for 3 hours in 10 mL glass flasks connected through Swagelok Ultra-Torr tee fittings to 330



mL glass flasks equipped with Louwers high-vacuum-valves. The flasks were flushed with $CO_2$-free air at
room temperature close to field conditions. The $CO_2$ was allowed to accumulate to at least 2000 ppm (~2
h).
The heterotrophic ($\delta^{13}C_h$) end-member was estimated as in Taylor et al. (2015), and similar to the root-
incubation experiment, soil samples from the top 5 cm of the litter layer or 10 cm below the soil surface
were collected, and roots were carefully removed to isolate heterotrophic components. Root-free soils were
placed in 10 mL glass flasks and allowed to incubate for 24 hours before being transferred to evacuated
330 mL glass flasks. The inorganic source ($\delta^{13}C_i$) end-member was estimated using one gram of dry soil
ground to pass through a 0.5 mm mesh placed in a 10 mL tube with a septum cap; then, 12 mL of 1M HCl
was added to dissolve the carbonate fraction, and the fumigated $CO_2$ withdrawn from each tube was
collected using a 10 mL syringe and injected into a 330- mL evacuated flask for isotopic analysis.
Radiocarbon estimates were based on the work of Carmi et al. (2013) at the same site, adjusted to the
measured atmospheric $^{14}C$ values during the study period (49.5 ‰; Carmi et al., 2013). The $\Delta^{14}C_{sa}$ and
$\Delta^{14}C_h$ end-members were estimated based on the assumption that they carry the $^{14}C$ signatures of 4 and 8.5
years, respectively, older than the $^{14}C$ signature of the atmosphere at the time of sampling, based on mean
ages previously estimated (Graven et al., 2012; Levin et al., 2010; Taylor et al., 2015). $\Delta^{14}C_i$ was obtained
from Carmi et al. (2013; Table 2). Monthly values of $\Delta^{14}C_{Rs}$ were obtained using the linear equation of the
regression line of the measured $\delta^{13}C$ values of Rsa, Rsh, and Ri and the corresponding estimated $\Delta^{14}C$
values (Fig. S2) and monthly $\delta^{13}C$ values of Rs.
**2.5. Isotopic analysis**
Isotopic analysis followed the methodology described in Hemming et al. (2005). The $\delta^{13}C$ of $CO_2$ in the air
was analyzed using a continuous flow mass spectrometer connected a 15-flask automatic manifold system.
An aliquot of 1.5 mL of air was expanded from each flask into a sampling loop on a 15-position valve
(Valco Houston, TX, USA). $CO_2$ was cryogenically trapped from the air samples using helium as a carrier
gas; it was then separated from $N_2O$ with Carbosieve G (Sigmaaldrich) packed column at 70°C and analyzed
on a Europa 20-20 Isotope Ratio Mass Spectrometer (Crewe, UK). $\delta^{13}C$ results were quoted in parts per
thousand (‰) relative to the VPDB international standard. The analytical precision was 0.1 %. To measure
$CO_2$, an additional 40.0 mL sub-sample of air from each flask was expanded into mechanical bellows and
passed through an infrared gas analyzer (LICOR 6262; Lincoln, NE, USA) in an automated system. The





precision of these measurements was 0.1 ppm. Flasks filled with calibrated standard air were measured with
each batch of 10 sample flasks; five standards were measured per 10 samples for $\delta^{13}C$ analyses and four
standards per 10 samples for $CO_2$ analyses.
Organic matter samples were dried at 60°C and milled using a Wiley Mill fitted with size 40 mesh, and soil
samples were ground in a pestle and mortar. Soils containing carbonates were treated with M hydrochloric
acid. Between 0.2 and 0.4 mg of each dry sample were weighed into tin capsules (Elemental Microanalysis
Ltd., Okehampton, UK), and their $\delta^{13}C$ was determined using an elemental analyzer linked to a Micromass
Optima IRMS (Manchester, UK). Three replicates of each sample were analyzed, and two samples of a
laboratory working standard cellulose were measured for every 12 samples. Four samples of the acetanilide
(Elemental Microanalysis Ltd.) international standard were used to calibrate each run, and a correction was
applied to account for the influence of a blank cup. The precision was 0.1‰.

### 2.6. Total below ground carbon allocation (TBCA)

TBCA (g C m$^{-2}$ y$^{-1}$) was calculated following Giardina and Ryan (2002) for the study year (2015–2016) as
follows,
$$\text{TBCA} = Rs - Rl + \Delta C_{soil} , \tag{12}$$
where $R_l$ is the annual above ground litter production from 2014–2015, and $\Delta C_{soil}$ is the annual change in
below ground total soil organic C. Litter production, not measured during the present study, was estimated
based on values obtained by Masyk et al. (2008b) for 2000–2006 (56 g C m$^{-2}$ y$^{-1}$) assumed to have increased
in the study period (2014–2015) proportionally to the measured increase in leaf area index (LAI; 1.31 to
1.9; i.e., $R_l = [(1.9*56)/1.31 = 83$ g C m$^{-2}$y$^{-1}$]) and herbaceous litter production. Three plots of 25 m$^2$ were
randomly selected in 2002 and harvested at the end of the growing season, at which time total fresh biomass
was weighed and subsamples were used to determine dry weight and C content. Grünzweig et al. (2007)
found that herbaceous litter production was close to the average rainfall for the specific year. This method
was adapted in the current study for the period between 2014 and 2015. Since above ground litter (Rl; the
sum of tree litter and herbaceous litter production) of a given year was mainly produced during that year
but decayed during the following hydrological year, TBCA was the current year's Rs and previous year's
Rl. $\Delta C_{soil}$ was set to constant as the average annual below ground carbon increase since afforestation
(Qubaja et al., unpublished).





**2.6. Statistical analyses**

A paired t-test was used to detect significant differences in Rs and metrological parameters between microsites (OA, BT, and UT) with the significance level set at 0.05. Pearson correlation analysis ($r$) was used to detect the correlation between Rs and meteorological parameters.

To quantify spatial-temporal variability in Rs, the coefficient of variation (CV %) was calculated as

$$CV = \frac{\text{Standard deviation}}{\text{mean}} \times 100\% \,. \tag{13}$$

Heterogeneity was considered weak if CV % $\leq$ 10 %, moderate if 10 % $<$ CV % $\leq$ 100 %, and strong if CV % $>$ 100 %. All the analyses were performed using Matlab software, version R2017b (MathWorks, Inc.).

**3. Results**

**3.1. Spatial variations**

The spatial variations in Rs across collars and microsites are reported in Table 1 together with other measured variables. The results indicated an overall mean Rs value of $0.8 \pm 0.1$ µmol m$^{-2}$ s$^{-1}$ with distinct values for the three microsites. Rs was greater at the UT site than at the BT and OA sites by a factor of ~2. The spatial variability among the microsites was also apparent in the Rs daily cycle (Fig. 1), with clear differences between the wet season (November to April), when the UT showed consistently higher Rs values than at other sites by a factor of about 1.6, and the dry season by a factor of about 2.6. Note that the daily peak in Rs remains at mid-day in both the wet and dry seasons. Overall, the 21 collars showed moderate variations (CV = 55 %; Table 1), negative correlations between Rs and distance from trees (Dt; $r$ = 0.6, $p < 0.05$), similar but statistically insignificant correlations between soil and air temperatures (Ts and Ta; $r$ = 0.3), and positive correlations between SWC and relative humidity (RH; $r$ = 0.3 and 0.2, respectively).



**Table 1 |** Annual mean of half-hour values across sites (OA, open area; BT, between trees; UT, under tree) and study period, of soil respiration flux rates (Rs)
together with the soil water content at 10 cm depth (SWC), minimum distances from nearby tree (Dt), soil temperature at 5 cm depth (Ts), and air temperature (Ta)
and relative humidity (RH) at the soil surface; (numbers in parenthesis indicate ±se).

| Sites | Points | Rs [$\mu mol\ m^{-2}\ s^{-1}$] | SWC [$x100\ m^3\ m^{-3}$] | Dt [m] | Ts [ºC] | Ta [ºC] | RH [%] |
|---|---|---|---|---|---|---|---|
| OA | 1_1 | 1.64 (0.02) | 16.5 (0.2) | 2.9 | 15.6 (0.1) | 15.4 (0.2) | 59.7 (0.5) |
| | 2_1 | 0.72 (0.01) | 14.5 (0.3) | 3.6 | 15.9 (0.2) | 15.0 (0.2) | 58.4 (0.6) |
| | 3_1 | 1.23 (0.02) | 19.3 (0.3) | 7.0 | 20.6 (0.3) | 18.2 (0.2) | 53.5 (0.5) |
| | 4_1 | 0.38 (0.01) | 11.3 (0.2) | 3.0 | 22.6 (0.2) | 20.8 (0.1) | 58.9 (0.4) |
| | 5_1 | 0.38 (0.01) | 5.8 (0.0) | 3.0 | 25.5 (0.1) | 24.0 (0.1) | 43.1 (0.4) |
| | 6_1 | 0.31 (0.01) | 5.7 (0.1) | 2.8 | 30.0 (0.3) | 26.2 (0.3) | 51.8 (0.9) |
| | 7_1 | 0.14 (0.01) | 6.1 (0.0) | 3.5 | 25.5 (0.2) | 23.2 (0.3) | 44.5 (0.9) |
| | **Average** | **0.68 (0.21)** | **11.3 (2.1)** | **3.7 (0.6)** | **22.3 (2.0)** | **20.4 (1.6)** | **52.8 (2.6)** |
| | **CV [%]** | **81 %** | **50 %** | **41 %** | **24 %** | **21 %** | **13 %** |
| BT | 1_2 | 0.77 (0.01) | 10.5 (0.1) | 1.8 | 16.1 (0.1) | 15.2 (0.2) | 60.5 (0.5) |
| | 2_2 | 0.88 (0.01) | 12.1 (0.2) | 1.5 | 14.8 (0.2) | 14.7 (0.2) | 59.5 (0.6) |
| | 3_2 | 0.84 (0.01) | 20.4 (0.3) | 2.7 | 20.1 (0.3) | 18.4 (0.2) | 54.1 (0.6) |
| | 4_2 | 0.91 (0.01) | 14.4 (0.2) | 2.7 | 23.3 (0.2) | 21.3 (0.2) | 58.5 (0.4) |
| | 5_2 | 0.41 (0.00) | 3.9 (0.0) | 2.0 | 24.6 (0.1) | 24.0 (0.1) | 43.2 (0.4) |
| | 6_2 | 0.41 (0.01) | 3.3 (0.1) | 2.5 | 29.1 (0.2) | 26.0 (0.3) | 52.5 (0.8) |
| | 7_2 | 0.46 (0.01) | 5.5 (0.0) | 1.2 | 23.9 (0.1) | 22.8 (0.3) | 45.7 (0.9) |
| | **Average** | **0.67 (0.09)** | **10.0 (2.4)** | **2.0 (0.2)** | **21.7 (1.9)** | **20.3 (1.6)** | **53.4 (2.6)** |
| | **CV [%]** | **35 %** | **63 %** | **29 %** | **23 %** | **21 %** | **13 %** |
| UT | 1_3 | 1.22 (0.02) | 9.3 (0.1) | 0.2 | 15.7 (0.1) | 15.2 (0.2) | 60.0 (0.5) |
| | 2_3 | 1.42 (0.01) | 14.0 (0.2) | 0.3 | 14.8 (0.2) | 14.8 (0.2) | 59.4 (0.6) |
| | 3_3 | 1.64 (0.01) | 19.8 (0.3) | 0.5 | 19.0 (0.2) | 18.0 (0.2) | 54.5 (0.6) |
| | 4_3 | 1.90 (0.02) | 11.3 (0.1) | 0.6 | 22.0 (0.1) | 20.8 (0.1) | 59.0 (0.4) |
| | 5_3 | 1.16 (0.01) | 4.0 (0.0) | 0.4 | 23.9 (0.1) | 23.7 (0.1) | 44.1 (0.4) |
| | 6_3 | 1.29 (0.01) | 4.5 (0.1) | 0.2 | 29.5 (0.3) | 25.9 (0.3) | 52.7 (0.9) |
| | 7_3 | 0.89 (0.01) | 5.2 (0.1) | 0.2 | 25.0 (0.1) | 23.0 (0.3) | 45.5 (0.9) |
| | **Average** | **1.45 (0.13)** | **9.7 (2.2)** | **0.3 (0.1)** | **21.4 (2.0)** | **20.2 (1.6)** | **53.6 (2.5)** |
| | **CV [%]** | **25 %** | **60 %** | **46 %** | **25 %** | **21 %** | **12 %** |
| All | **Average (SE)** | **0.8 (0.1)** | **10 (1)** | **2.0 (0.4)** | **21.8 (1.1)** | **20.3 (0.9)** | **53.3 (1.4)** |
| | **Max** | **1.90** | **20** | **7.0** | **30.0** | **26.2** | **60.5** |
| | **Min** | **0.14** | **3** | **0.2** | **14.8** | **14.7** | **43.1** |
| | **CV [%]** | **55 %** | **55 %** | **82 %** | **23 %** | **20 %** | **12 %** |



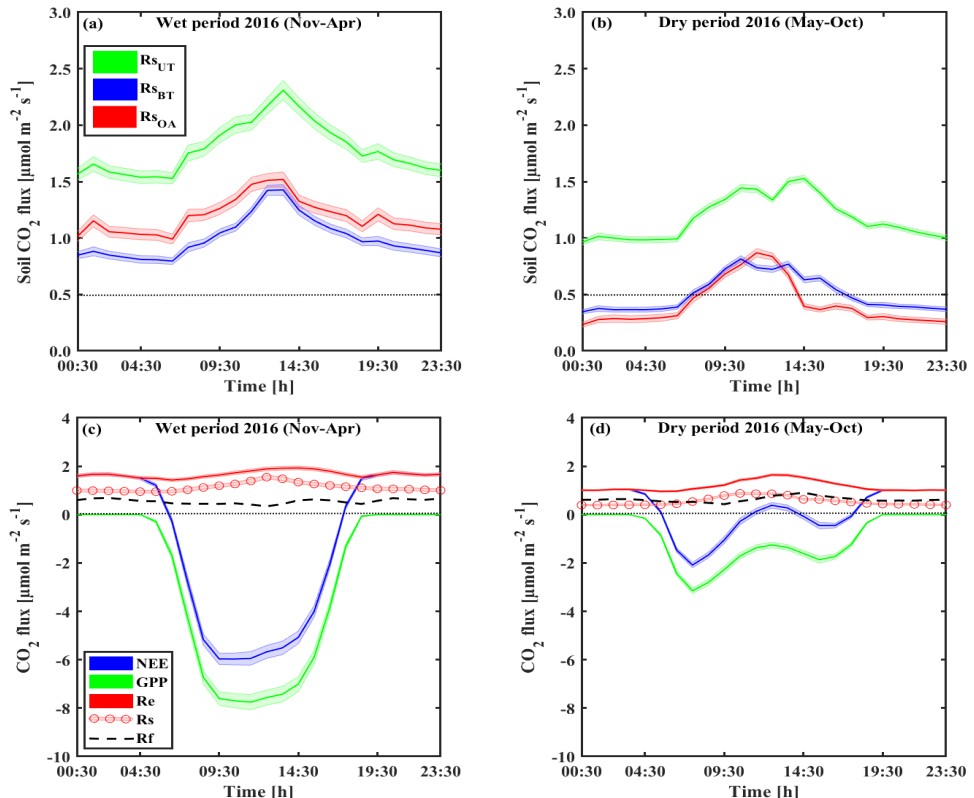

**Figure 1 |** Representative diurnal cycles of soil respiration (Rs; using soil chambers across microsites: open-area, OA; between-trees, BT; under-trees, UT) in panels a and b, and of net ecosystem exchange (NEE; canopy scale eddy covariance) and gross primary production (GPP), and ecosystem respiration (Re) and its partitioning to soil respiration (Rs) and aboveground tree respiration (Rf) in panels c and d, during the wet (Nov-Apr) and dry (May-Oct) periods. Based on half-hour values over the diurnal cycle; shaded areas indicate ±se; Rf was estimated as the residual as Rf=Re-Rs and was presented as black-dashed line.

**3.2. Temporal dynamics**

On the diurnal timescale, $CO_2$ fluxes showed typical daily cycles (Fig. 1). As expected, on average, all $CO_2$ fluxes were higher during the wet period compared to the dry season by a factor of ~2. However, Rs and Re peaked around mid-day in both the wet and dry seasons, while the more physiologically controlled NEE and GPP showed a shift from mid-day (around 11:00–14:00) to early morning (08:00–11:00) in the dry season, with a mid-day depression and a secondary afternoon peak (Fig. 1d).

The temporal variations across the seasonal cycle are reported in Fig. 2 based on monthly mean values, exhibiting sharp differences between the wet and dry seasons. As previously observed in this semi-arid site, all $CO_2$ fluxes peak in early spring between March and April. The corresponding high-resolution data

are reported in Fig. S5 and show that in high winter (February), Rs rates were associated with clear days
when photosynthetic active radiation (PAR) increased with air temperature Ta. These data also show that
following rainy days, daily Rs values could reach 6.1 µmol m$^{-2}$ s$^{-1}$, although the average was 1.1 ± 0.2
µmol m$^{-2}$ s$^{-1}$ during the wet period, which diminished by ~55 % in the dry season to mean daily values of
0.5 ± 0.1 µmol m$^{-2}$ s$^{-1}$. In spring (April), all CO$_2$ fluxes peaked during the crossover trends of decreasing
soil moisture content, and increasing temperature, and PAR (Fig. S5).

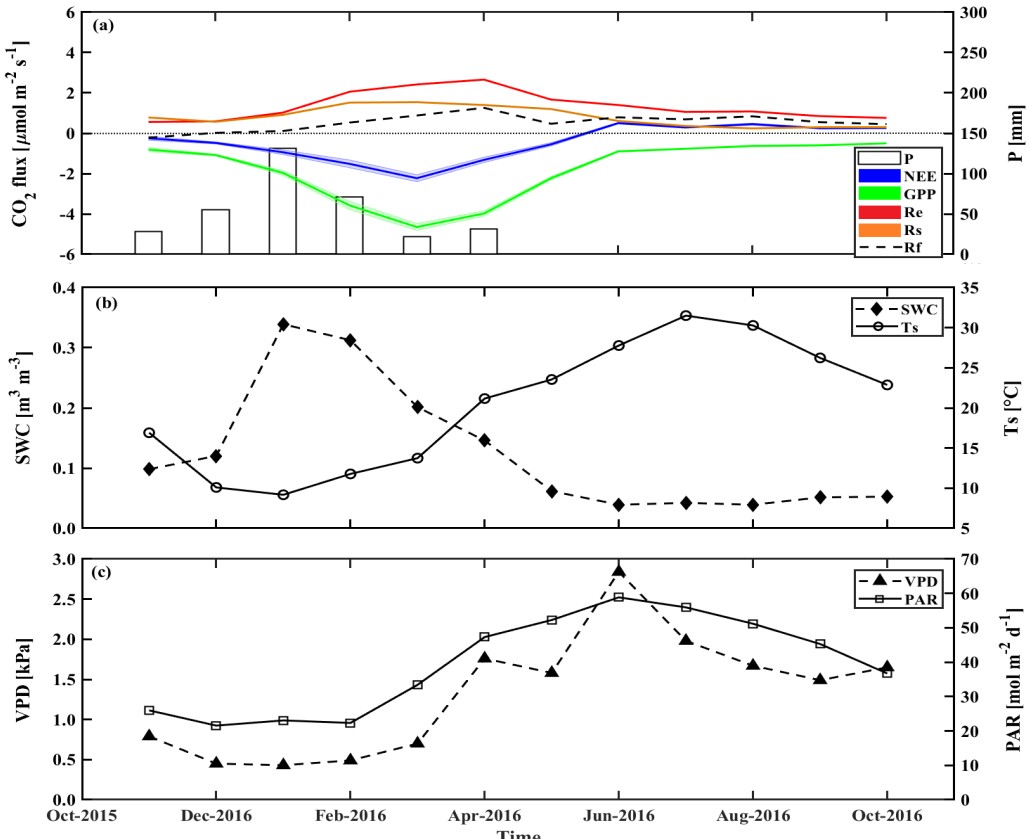


**Figure 2|** Seasonal trends of monthly mean values during the research period of a) the fluxes of net ecosystem
exchange (NEE), gross primary production (GPP), and ecosystem respiration (Re) and its components soil respiration
(Rs) and aboveground tree respiration (Rf); and monthly mean of key environmental parameters, b) soil water content
at the top 10 cm (SWC$_{0-10}$) and soil temperature at 5 cm (Ts), and c) vapor pressure deficit (VPD) and photosynthetic
activity radiation (PAR). Rf is obtained from the Re-Rs.


The temporal variations in the half-hour values of Rs reflected changes in soil moisture at 0–5 cm depth
and PAR ($r$ = 0.5 and 0.2, respectively; $p < 0.01$) and negative correlations with Ts and RH ($r$ = 0.2 and
0.1, respectively; $p < 0.01$). The variations in the integrated Rs showed a CV of 71%, with the temporal




variations dominated by PAR (CV > 100 %) and moderately affected by SWC (CV ~ 85 %), Ts, and RH
(CV ~ 40 %). Repeating the models applied by Grünzweig et al. (2009), the potential climatic factors that
best predicted daily Rs shifted from SWC and PAR in the dry season to Ts and PAR in the wet season
(Table S2). These equations explained 43 % and 70 % of the variation in Rs in the dry and wet seasons,
respectively (Table S2). A reasonable forecast of the temporal variations in Rs at half-hour values ($R^2$ =
0.60, $p < 0.0001$) were obtained based on $SWC_{0-10}$ and Ts values across the entire seasonal cycle based on:

$$Rs(\mu mol\ m^{-2}\ s^{-1}) = 0.05126 * \exp(0.04274 * Ts + 28.51 * SWC - 74.44 * SWC^2). \quad (14)$$

At the ecosystem scale, Re was characterized by high fluxes in the wet season and peak values of ~2.4
$\mu mol\ m^{-2}\ s^{-1}$ in February to April (Fig. 2; Table S1). Refluxes rapidly decreased after rainy cessation and
reached the lowest values in the fall (SeptemberOctober) with mean dry period values of $0.5 \pm 0.1$ µmol
$m^{-2}\ s^{-1}$ (Fig. 2, Table S1). GPP had a mean value of $-1.8 \pm 0.4$ µmol $m^{-2}\ s^{-1}$, and daily NEE had a mean
value of $-0.5 \pm 0.3$ µmol $m^{-2}\ s^{-1}$ (Table S1 and Fig. S5), with the same seasonality (Fig. 2).
**Table 2** | $\delta^{13}$C and $\Delta^{14}$C signature of soil respiration (Rs) and its partitioning to autotrophic (Rsa), heterotrophic (Rh),
and the abiotic (Ri), together with the relative contribution of each to the soil and ecosystem respiration for Yatir
forest during 8 campaigns of measurements from January to September 2016 (numbers in parenthesis indicate ±se).
The monthly contribution of Rsa, Rh, and Ri to Rs or Re are presented in Fig. 3a and b, respectively.

| Signature | Rsa | Rh | Ri | Rs |
|---|---|---|---|---|
| | [‰] | | | |
| $\delta^{13}$C | -23.7 (0.5)[1] | -24.3 (0.0)[1] | -6.5 (0.0)[1] | -20.8 (±0.6)[1] |
| $\Delta^{14}$C | 30[3] | 50[3] | -900[2] | -134 (34)[4] |
| **Relative contribution to Rs** | **0.40 (0.02)** | **0.39 (0.02)** | **0.21 (0.04)** | |
| **Relative contribution to Re** | **0.24 (0.04)** | **0.23 (0.04)** | **0.13 (0.01)** | **0.60 (0.06)** |

[1] Measured at the present study.
[2] Measured by Carmi et al., (2013).
[3] Calculated based on the measured atmospheric value by Carmi et al., (2013).
and [4] Calculated based on the best fit regreeion equation in Figure S2.

Figure 3 (see also Table 2) summarizes the seasonal variations in Rs and Re partitioning. The monthly Rsa
and Rh were not significantly different but were significantly different from Ri ($p < 0.05$). The Rsa/Rs
ratios ranged from 0.32 to 0.46, with the largest contribution in early spring from Februaryto–April. The
Rh/Rs fraction ranged between 0.33 and 0.45, being highest during the wet season. The Ri/Rs fraction of
total respiration from inorganic sources ranged from 0.09 to 0.35, with the largest contribution in the driest
period. The mean relative contribution of these components to Rs over the sampling campaigns are
presented in Figure 3a, but on average, soil biotic fluxes were higher than abiotic fluxes by a factor of ~4.
Repartitioning showed an average increase in Rf/Re from 25 % in the wet season to 54 % in the dry season
and a decline in Rs/Re from 75 % to 46 % on average in the wet and the dry seasons, which reflected a



seasonal change of Rf in the wet season to peak values in the dry season (Fig. 3b). Both the highest and
lowest Rs fractions (~0.74 and nearly 0.34) along the seasonal cycle were associated with low total Re
fluxes, that is in the fall before the Rf peak in the spring and in the summer when physiological controls
limited water loss.

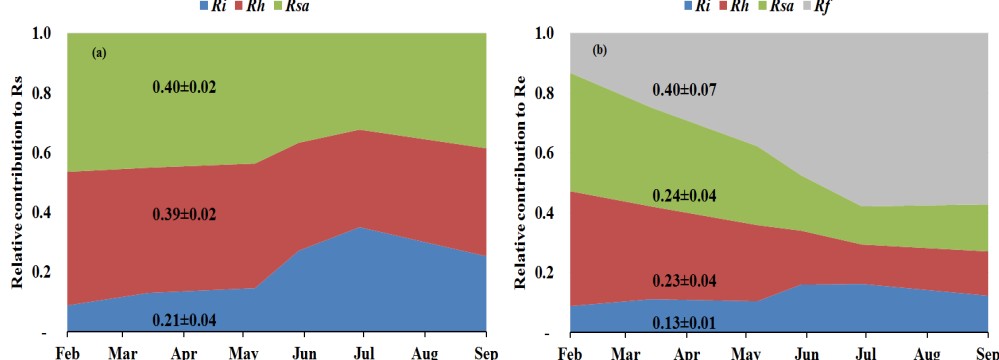


**Figure 3 |** a) Linear mixing models $\delta^{13}C$ and $\Delta^{14}C$ of soil respiration (Rs) isotope signatures (from soil $CO_2$ profile
method at 0, 30, 60, 90, and 120 cm soil depth) were used to determine the seasonal variations in the relative
contribution of soil autotrophic (Rsa), heterotrophic (Rh), and abiotic (Ri) components to Rs, and b) seasonal
variations in the relative contribution of soil autotrophic (Rsa), heterotrophic (Rh), abiotic (Ri), and foliage and stem
respiration (Rf is obtained from the Re-Rs.) components to ecosystem respiration (Re) during 8 campaigns (Jan-Sep)
in 2016. These results confirmed earlier estimates of Grünzweig et al. (2009) and Maseyk et al. (2008a).
**3.4. Annual scale**
**Table 3 |** Mean annual values of ecosystem respiration (Re), its components and associated ratios, net ecosystem
exchange (NEE; from eddy covariance), net primary productivity (NPP), gross primary productivity (GPP), carbon-
use efficiency (CUE), leaf aria index (LAI), and ratio of total belowground Carbon allocation (TBCA) to GPP
(TBCA/GPP) in the present study (mean of Nov-2015 to Oct-2016) and in comparison to results obtained previously
at the same site (2001-2006 mean values). Ri, Rh, Rsa, Rs, Rl and Rw denote abiotic, heterotrophic, autotrophic, soil,
foliage, and wood $CO_2$ flux, $\Delta$ indicates the difference between the mean values for the two studies; using mean
values for the two study periods. $Q_{10}$ as derived during the two studies for the wet and dry season.

| Study | Ri | Rh | Rsa | Rs | Rl | Rw [g m$^{-2}$ y$^{-1}$] | Re | NEE | NPP | GPP |
|---|---|---|---|---|---|---|---|---|---|---|
| **Mean (2001-2006)** | 56 | 139 | 211 | 406 | 260 | 70 | 735 | -211 | 350 | -880 |
| **x/Re** | 0.08 | 0.19 | 0.29 | 0.55 | 0.35 | 0.10 | | | | |
| **x/GPP** | 0.06 | 0.16 | 0.24 | 0.46 | 0.30 | 0.08 | 0.84 | 0.24 | 0.40 | |
| | | | | | | | | | | |
| **Mean (2015-2016)** | 61 | 115 | 119 | 295 | 156 | 39 | 488 | -167 | 276 | -655 |
| **x/Re** | 0.13 | 0.23 | 0.24 | 0.60 | 0.32 | 0.08 | | | | |
| **x/GPP** | 0.09 | 0.18 | 0.18 | 0.45 | 0.24 | 0.06 | 0.75 | 0.25 | 0.42 | |


| Study | CUE | $Q_{10}$ SWC$^1$ | SWC$^2$ | LAI [m$^2$ m$^{-2}$] | TBCA/GPP | $\Delta$Rf-ratio/$\Delta$LAI | $\Delta$Rs-ratio/$\Delta$LAI |
|---|---|---|---|---|---|---|---|
| **Mean (2001-2006)** | 0.40 | 2.5 | 1.2 | 1.3 | 0.41 | | |
| | | | | | | -0.05 | 0.07 |
| **Mean (2015-2016)** | 0.42 | 1.6 | 1.1 | 2.1 | 0.38 | | |

$^1$ SWC$\geq$0.2 [m$^3$ m$^{-3}$].and $^2$ SWC<0.2 [m$^3$ m$^{-3}$].





On an annual time-scale, estimates of $CO_2$ flux components based on EC measurements resulted in annual
values of GPP, NPP, Re, and NEP of 655, 276, 488, and 167 g C $m^{-2}$ $y^{-1}$, respectively (Tables 3 and S1).
On average across the measurement period, Rs was the main $CO_2$ flux into atmosphere, making up 60 ±
6% of Re (295 ± 4 g C $m^{-2}$ $y^{-1}$; Tables 3 and S1), and Rf was another significant component accounting for
40 ± 6% of Re (Fig. 3b), which reflected the low density (300 trees $ha^{-1}$) nature of the semi-arid forest. As
indicated above, Re partitioning showed a decrease in Rs/Re and an increase in Rf/Re from winter to
summer, which is clearly apparent in Fig. 3b. On an annual scale, during the study period, estimates of Rf,
Rsa, Rh, and Ri values were 194 ± 36, 119 ± 21, 115 ± 20, and 61 ± 6 g C $m^{-2}$ $y^{-1}$, respectively. Despite
relatively high rates of respiration fluxes, the CUE of the ecosystem remained high at 0.42.
**Table 4 |** Linear regression over time of ecosystem respiration (Re), its components and associated ratios, net
ecosystem exchange (NEE; from eddy covariance), net primary productivity (NPP), gross primary productivity
(GPP), total belowground Carbon allocation (TBCA), and ratio of TBCA/GPP (2001-2016) in the semi-arid pine
forest site. Rh, Rs, and Rf denote heterotrophic, soil, and foliage and wood $CO_2$ flux. The model parameters (slope
and intercept) reported for each variable together with the squared coefficient of regression ($R^2$) and the *P-value*. The
data for the regression analysis includes 15 years (2001-2016) of the site's flux records of Re, NEE, GPP, NPP; and
from comparing the data from the earlier 2001-2006 to the present stud (Nov-2015-Oct-2016) for 7 years of Rs, Rh,
Rf, and TBCA and TBCA/GPP.

| Variables | Linear regression | | | |
| --- | --- | --- | --- | --- |
| | **Slope** | **Intercept** | **$R^2$** | ***P-value*** |
| **Re** | 1.0 | 386 | 0.0 | 0.819 |
| **Rs** | **-11.0** | **436** | **0.6** | **0.036** |
| **Rh** | -3.9 | 159 | 0.1 | 0.404 |
| **Rf** | -7.1 | 341 | 0.3 | 0.180 |
| **NEE** | 9.0 | -231 | 0.2 | 0.065 |
| **GPP** | 8.0 | -617 | 0.1 | 0.254 |
| **NPP** | -8.8 | 304 | 0.2 | 0.083 |
| **TBCA** | -9.4 | 367 | 0.5 | 0.114 |
| **TBCA/GPP** | -0.006 | 0.43 | 0.2 | 0.416 |


Using the site records of nearly 20 years, long-term trends in GPP, NPP, Re, and NEP were obtained.
Comparison of present results with the 2001-2006 values obtained by Grünzweig et al. (2009) and Maseyk
et al. (2008a) provided a basis for estimating the temporal trends in soil respiration. Notably, no clear or
significant trend over time was observed in any of the canopy-scale fluxes (Table 4). However, the soil
respiration rates, Rs, decreased significantly by 11 ± 4 g C $m^{-2}$ $y^{-1}$ ($R^2 = 0.62$, $p = 0.036$; Table 4) from
regression of the 2001-2006 value to the mean value in the present study (Nov-2015-Oct-2016). Because
of strong influenced of variations in environmental conditions, primarily precipitation on interannual
variations, it is likely that the relative contributions of the different fluxes, expressed as ratios in Table 3,
provide a relatively robust perspective of the long-term temporal changes in the ecosystem functioning.
Based on long-term results presented in Table 3, Rf/Re (Rf ratio) decreased by 11 % while Rs/Re (Rs ratio)



increased by 9 % over the observation period noted above (essentially 2003 to 2016). Notably, LAI
increased across the same observation period from 1.3 to 2.1 (+57 %; Qubaja et al., unpublished), indicating
that the ΔRf-ratio/ΔLAI decreased (-0.05/0.75=-0.07), and the ΔRs-ratio/ΔLAI increased
(+0.05/0.75=+0.07; Table 3). These results highlight the shift from Rf to Rs over the 13 years observation
period. Total TBCA was 247 g C m$^{-2}$ y$^{-1}$ during the study period and relative below ground partitioning of
carbon in the ecosystem (TBCA/GPP) averaged 38%, which was lower than the previous study by 7 % for
the same site (Grünzweig et al., 2009).
**4. Discussion**
Partitioning ecosystem carbon fluxes and long-term observational studies are key to understanding
ecosystem carbon dynamics and their response to change. This research hypothesized that soil $CO_2$ efflux
at the dry study site is a key factor underlying the observation of high NEP and high CUE in this system.
Comparing $CO_2$ fluxes in this forest with fluxes in a range of European forests showed that mean NEP in
the semi-arid forest (160 g C m$^{-2}$ y$^{-1}$) was similar to the mean NEP across other European forests (150 g C
m$^{-2}$ y$^{-1}$; FLUXNET). In contrast, soil $CO_2$ efflux in the semi-arid forest was 295 g C m$^{-2}$ y$^{-1}$, which is at the
low end of Rs values across the range for other climatic regions, from 50 to 2750 g C m$^{-2}$ y$^{-1}$ (Adachi et al.,
2017; Bond-Lamberty, 2018; Chen et al., 2014; Grünzweig et al., 2009; Hashimoto et al., 2015). This is
clearly lower than the mean Rs value for global evergreen needle forests, which is estimated at 690 g C m$^{-}$
$^2$ y$^{-1}$ (Chen et al., 2014), and between estimates for desert scrub and Mediterranean woodland (224–713 g
C m$^{-2}$ y$^{-1}$; Raich and Schelsinger, 1992). The mean rate of Rs, 0.8 µmol m$^{-2}$ s$^{-1}$, is also in the range reported
for unmanaged forest and grassland in the dry Mediterranean region (0.5 and 2.1 µmol m$^{-2}$ s$^{-1}$; Correia et
al., 2012). High productivity associated with low Rs supports the high CUE estimate and the carbon
sequestration potential of the semi-arid Aleppo pin plantation (Rotenberg and Yakir, 2010).
The spatial variations in Rs among the microsites (Table 1) can be approximated to estimate the possible
impact on Rs caused by changes in forest density associated with a drying climate (e.g., tree thinning and
motality). For example, decreasing from the present optimal stand density (Raz-Yaseef et al., 2010) of 300
trees ha$^{-1}$ to 100 trees ha$^{-1}$ would result, based on present results, in decreasing ecosystem Rs and increasing
soil evaporation (Es) by 11 % and 38 %, respectively. This is consistent with the trend for increasing stand
density (Litton et al., 2004) and the effects of stand density on Rs through effected litterfall, TBCA, and
total soil N (Noh et al., 2010). More work is needed to determine the net effects of such changes (see
Maseyk et al., 2011; Villegas et al., 2015).
Low Rs values were associated with relatively high rates of autotrophic respiration, with a mean annual-
scale Rsa/Rs ratio of 0.40, which is similar to the estimated global mean value within the range of 0.09 to
0.49 (Chen et al., 2014; Hashimoto et al., 2015). Heterotrophic respiration was of the same magnitude



(annual-scale Rh/Rs ratio: 0.39 ± 0.02; Table 2 and Fig. 3), which reflected an increasing trend of +14 %
from mean 2001-2006 values (Grünzweig et al., 2009; Maseyk et al., 2008a) to the mean values in the
present study. This ratio is lower than the estimated global mean Rh/Rs of 0.56 (Hashimoto et al., 2015)
but comparable to the long-term trend of +17 % between 1990 and 2014 (Bond-Lamberty et al., 2018).
The seasonal dynamics in Rs partitioning (Fig. 3) might reflect an increasing balance between Rsa and Rh
compared to Rsa dominance in the dry season, which was present in the earlier study (Grünzweig et al.,
2009; Maseyk et al., 2008a; see also Carbone et al., 2008, Tang et al., 2005).
Carbon partitioning below ground (TBCA/GPP) was relatively high at 0.38, despite a ~7 % decrease in
values compared to those reported by Grünzweig et al. (2009) but was similar to the mean value for forests
in various biomes (Litton et al., 2007). This explains the high rate of SOC accumulation observed over the
period since afforestation (Grünzweig et al., 2007; Qubaja et al., unpublished). The ratio of Rs/GPP did
not change over the observation period (0.46 vs. 0.45 for 2001-2006 vs. 2015-2016 mean values) but shifted
to a larger contribution of Rh (Table 3).
The relatively low annual scale of the heterotrophic respiration to Rs was consistent with the dry surface
soil layer over much of the year in this forest (Figures 2 and S5) and the observed low decomposability of
plant detritus and high mean SOC accumulation rate (Grünzweig et al., 2007). The small increase in Rh
proportional contribution to Rs may reflect the general climatic trends in the region of increasing
temperature, with no significant change in precipitation (Bond-Lamberty et al., 2018).
The relatively low Rs under conditions of high temperature in the semi-arid ecosystem implies reduced to
sensitivity of respiration to temperature. This is partly imposed by low SWC conditions during extended
parts of the year (Grünzweig et al., 2009; cf. Rey et al., 2002; Xu and Qi, 2001). Accordingly, Rs varied
with Ts only under relatively wet conditions, but during 8–9 months of the year when SWC was below a
threshold value of 0.2 $m^3$ $m^{-3}$, Rs varied with water availability. The Rs vs. Ts relationship used to estimate
apparent $Q_{10}$ values indicated a decrease from $Q_{10} = 1.6$ in the wet season to $Q_{10} = 1.1$ in the dry season.
This represents a similar trend in $Q_{10}$ compared to the results of Grünzweig et al. (2009; Table 3). The
estimated $Q_{10}$ values are consistent with published values that range from 1.4 to 2.0 among different
ecosystems (Hashimoto et al., 2015; Zhou et al., 2009) and with low values under low SWC (Reichstein et
al., 2003; Tang et al., 2005). The low temperature sensitivity in the dry season must be related to reduced
microbial activity but may also involve down regulation of the plant activity (Maseyk et al., 2008a) and
drought-induced dormancy of shallow roots (Schiller, 2000). Little is known about the differences in $Q_{10}$
of soil Rh and Rsa respirations (Yu et al., 2017), which could respond differently to different environmental
variables (Matteucci et al., 2015) and make distinct contributions to soil carbon sequestration (Kuzyakov,
2006). Previous studies have shown that Rh and Rsa exhibit different temperature sensitivities (Rey et al.,





2002), but the underlying causes of the temporal changes observed in the $Q_{10}$ values at the study sites in
this study remain uncertain.
Dry lands can obtain water from sources other than precipitation, yet little is known about how non-rainfall
water influences dry land communities and activity. The results presented here and in previous studies
(Agam and Berliner, 2004; Kosmas et al., 1998) show that water vapor adsorption may occur relatively
frequently in dry seasons in dry environments. Re-evaporation is proposed to help protect against the
effects of extreme drought and to influence other ecosystem processes, such as stimulating microbial
activities, litter decomposition, and desert lichen activity (Dirks et al., 2010; Gliksman et al., 2017; Kuehn
et al., 2004; Newell et al., 1985; Kappen et al., 1979; Lange et al., 2006). More information on the potential
role and importance of water-vapor adsorption in a dry environment is needed.
Net ecosystem $CO_2$ exchange measurements are usually assumed to be strongly dominated by biological
processes (e.g., photosynthesis, respiration, or microbial activities), while the contribution of nonbiological
processes, such as those associated with carbonate precipitation and dissolution reactions in calcareous
soils, are seldom considered. The relative importance of abiotic component to the $CO_2$ flux, however,
greatly increases in dry environments and in dry seasons when biological activities drastically decrease
(Kowalski et al., 2008; Lopez-Ballesteros et al., 2017; Serrano-Ortiz et al., 2010), which includes the
observation of enhanced weathering at the present study sites (Roland et al., 2012). Notably, ambiguity
exists among the terms 'source' and 'sink' when considering the $CO_2$ exchange between carbonate rocks
and the atmosphere and carbon sequestration (Eshel et al., 2007). In the present study, carbonate
precipitation was on average 9 % of the GPP and 13 % of Re during the study period (Table 3). This
demonstrates the importance of this flux in semi-arid systems and its potential influence on estimating
short-term Re fluxes (Angert et al., 2015; Roland et al., 2012).
*Data availability*
The data used in this study are archived and available from the corresponding author upon request
(dan.yakir@weizmann.ac.il).
*Author contributions*
RQ and DY designed the study; RQ, FT, ER and DY performed the experiments. RQ and DY analyzed the
data. DY and RQ wrote the paper with discussions and contributions to interpretations of the results from
all coauthors.
*Competing interests*



The authors declare that they have no conflict of interest.
**5. Acknowledgements**
This long-term study was funded by Forestry department of Keren-Kayemeth-LeIsrael (KKL) and the
German Research Foundation (DFG) as part of the project "Climate feedbacks and benefits of semi-arid
forests" (CliFF) and by the Israel Ministry of Science and the Ministry of National Education, Higher
Education, and Research (MENESR) of France (IMOS-French Program: 3-6735). The authors thank Efrat
Schwartz for assistance with lab work. The long-term operation of the Yatir Forest Research Field Site is
supported by the Cathy Wills and Robert Lewis Program in Environmental Science. We thank the entire
Yatir team for technical support and the local KKL personnel for their cooperation.

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

Biogeosciences, 114.