# Peer review of "Partitioning of canopy and soil CO2 fluxes in a pine forest at the dry"

_Biogeosciences, 2019_

## Referee Comment (RC1) · Anonymous Referee #1 · 26 Aug 2019

This paper uses soil and canopy fluxes with stable isotopes (13C) and radiocarbon (14C) measurements of a 50-year-old dry pine forest over one year to partition the ecosystem's CO2 flux into gross primary productivity (GPP) and ecosystem respiration (Re) and soil respiration flux into autotrophi (Rsa), heterotrophic (Rh), and inorganic (Ri) components. The measurements and data are valuable. The topics are be of great interest. But the writing is very confusing. Abstract: The abstract lacks critical information. Introduction: The introduction should be rephrased. There are too much pieces of information on general knowledge. The Introduction needs a better flow. The scientific significance should be addressed more. A description that explain why the paper is needed following the previous studies (i.e. the 2001-2006 study) would be very informative for the readers. Furthermore, I suggest the authors cite more relevant

papers on the Mediterranean climate zones and add one or two hypothesis. Site description: provide more information about the vegetation (e.g. root depth). Flux and meteorological measurements: How did the gaps in NEE and GPP are filled? How many missing data points are there due to instrument failure and quality control? Soil $CO_2$ flux: I'm really confused. How many data did the author used in the paper? Just one year? Using just one-year measurements can not identify the long-term temporal changes in the soil–atmosphere $CO_2$ fluxes in this environment. Discussion: The hypothesis should also be into Introduction. The present study used only one-year data, I suggest the authors using a tempered tone in the discussion part. The Discussion has the same problem with Introduction. There are many pieces of interesting information. But the discussion should be centered on several key aspects of your results. The Discussion should echoes the Introduction. I suggest rephrase the Discussion and make a better flow in the Discussion.

---

## Referee Comment (RC2) · Anonymous Referee #2 · 27 Aug 2019

This manuscript describes the study partitioning of canopy and soil CO2 fluxes in a pine forest at the dry timberline using the measurements of isotopic signatures ($\delta$13C and $\Delta$14C) of CO2 emitted from bulk soils, fine roots, root-free soils, and carbonate fractions. The measurement and data are interesting. Then, scientific insights, which can be gained from this study, would significantly contribute for improving our understanding the response of dry environment ecosystems to climate change. The writing, however, should be improved more and more as pointed out by Referee #1. Then, please refine every sentence in the manuscript more carefully, because there are substantial typos (e.g. "a pine forests" in the title, "Soil respiration from the atmosphere" in Line 29-30, "Reflux" in Line 369, and so on). In addition to these concerns for writing, I have a technical concern about the estimating $\delta$13C for CO2 emitted from bulk soils

(i.e. $\delta$13CRS in the manuscript). The authors estimated $\delta$13CRS using the keeling plots for soil CO2 profile data at 0, 30, 60, 90, and 120 cm depth; however, the $\delta$13C of soil organic matters, the major source of heterotrophic respiration, often change along with soil depth increase. Then, these vertical changes in $\delta$13C of soil organic matters have significant potentials affecting the $\delta$13C-CO2 profile. This means that the observed relationships between $\delta$13C-CO2 and CO2 concentration profiles might be affected not only by the change in contribution of source CO2 and background CO2, but also by the changes in $\delta$13C of source CO2. Therefore, in my opinion, the authors are needed to provide the reliable justification for their methodology, to quantify the uncertainty for estimated $\delta$13CRS, and/or to apply alternative methodology for estimating $\delta$13CRS. Finally, please consider to include the photographs showing conditions of each chamber site and the schematic diagrams describing three collars locations within a chamber site.

---

## Author Comment (AC1) · 30 Sep 2019

Detail response to two Reviewers comments, BG-2019-291

Ref1

This paper uses soil and canopy fluxes with stable isotopes (13C) and radiocarbon(14C) measurements of a 50-year-old dry pine forest over one year to partition the ecosystem's CO2 flux into gross primary productivity (GPP) and ecosystem respiration(Re) and soil respiration flux into autotrophi (Rsa), heterotrophic (Rh), and inorganic(Ri) components. The measurements and data are valuable. The topics are be of great interest.

Response: Thank you, it is the most important point. But as we clarify better in the

revisions (see also below) the paper goes beyond "one year of partitioning" as this year of measurements allowed us to combine it with our own study about 10 years earlier at the same site to provide a longer-term perspective on the changes in such partitioning (and as the saying goes, with the "whole is greater than the sum of its parts" . . .)

But the writing is very confusing.

Response: We are sorry this is the case and made a serious effort to fix this by better streamlining the paper (as also noted below), careful proof-reading and, considering that the authors are not native English speakers, also sending it out to professional editing.

Abstract: The abstract lacks critical information.

Response: We are not sure what is the missing critical information. We added in the Abstract key information on the site and the study itself. We concluded that this is likely the missing information after we carefully checked the summary in the Abstract and the Result. The results are presented in 3 figures and 4 tables, and carefully comparing the detail summary of the results in the Abstract, indicates that all the main points, including our observed values of GPP, Re, NPP, CUE, the partitioning (the main effort in this study) of respiration Rs to Rsa to Rh, Ri, the seasonal changes, the main long-term changes when combined with our earlier study, and the main occlusions.

Introduction: The introduction should be rephrased. There are too much pieces of information on general knowledge. The Introduction needs a better flow. The scientific significance should be addressed more. A description that explain why the paper is needed following the previous studies (i.e. the 2001-2006 study) would be very informative for the readers. Furthermore, I suggest the authors cite more relevant papers on the Mediterranean climate zones and add one or two hypothesis.

Response: Done. We shortened the introduction by nearly 30%. The original Intro

had 6 paragraphs, and we removed two non-essential ones, and combined the one on isotopes into the paragraph on partitioning. We also improved the Motivation (the importance of the combination of the two studies at our site in 2001-6 and 2015-16 to assess changes over time) and we provide clearer working hypotheses, as requested. We checked the literature and added missing references on Mediterranean studies.

Site description: provide more information about the vegetation (e.g. root depth). Flux and meteorological measurements: How did the gaps in NEE and GPP are filled? How many missing data points are there due to instrument failure and quality control?

Response: Done. We added the requested information, including detail on root depth and on the vegetation at the site (overstory and understory, indicating the main species). Specifically, regarding the requested information root depth we note that our paper published earlier this year (Preisler et al. 2019, Functional Ecology; cited) provides detail information on root depth, distribution and microsite effects (see SI Fig. 1).

Soil $CO_2$ flux: I'm really confused. How many data did the author used in the paper? Just one year? Using just one-year measurements can not identify the long-term temporal changes in the soil–atmosphere $CO_2$ fluxes in this environment.

Response: As noted above, we now clarify this issue straight-out in the Intro and again in the Methods and in the Discussion. Briefly, note that all the figures and tables indicate that the new data were obtained during 2015-2016 (one full year). But both the figures/tables and Discussion show that impotent virtue of the paper is in combining these new data with our earlier study at the same site (2001-2006) looking at the same parameters and obtaining a long-term perspective of the change in the flux component (i.e. in the partitioning) over a time window of about 10 years. Many of the studies reported in this journal aim at assessing change, especially in response to global change. While some parameters are monitored continuously (our flux tower operated continuously for 20 years), other measurements required for the Partitioning of soil

fluxes cannot be, practically, made continuously and here we combine the continuous measurement with the periodical campaign to assess sufficiently long-term changes.

Discussion: The hypothesis should also be into Introduction. The present study used only one-year data, I suggest the authors using a tempered tone in the discussion part. The Discussion has the same problem with Introduction. There are many pieces of interesting information. But the discussion should be centered on several key aspects of your results. The Discussion should echoes the Introduction. I suggest rephrase the Discussion and make a better flow in the Discussion.

Response: Done. The working Hypotheses are included in both the Introduction and Discussion as noted above, and we streamlined the Discussion to improve the correspondence with the Introduction. As also noted above, we better focus on the link of the new one-year data of this study with our older study at the same site to assess changes over a period of about 10 years to addresses the issue of changes in the ecosystem and particularly in the soil carbon flux component as climate is changing (we also added in the SI the long-term records of temperature and precipitations).

Ref2

This manuscript describes the study partitioning of canopy and soil $CO_2$ fluxes in a pine forest at the dry timberline using the measurements of isotopic signatures ($\delta$13Cand$\Delta$14C) of $CO_2$ emitted from bulk soils, fine roots, root-free soils, and carbonate fractions. The measurement and data are interesting. Then, scientific insights, which can be gained from this study, would significantly contribute for improving our under-standing the response of dry environment ecosystems to climate change.

Response: Thank you. It is important to see that this study is recognized as adding to understanding ecosystem response to change.

The writing, however, should be improved more and more as pointed out by Referee #1. Then, please refine every sentence in the manuscript more carefully, because

there are substantial typos (e.g. "a pine forests" in the title, "Soil respiration from the atmosphere" in Line 29-30, "Reflux" in Line 369, and so on).

Response: We recognize that our failure to submit an appropriately proofed manuscript made some significant damage, although there were no errors in the science. This was the result of some unfortunate confusion in combining the different versions of the paper proof-read by different coauthors. It has been fixed and considering that the authors are not native English speakers the paper has been sent out for professional editing.

In addition to these concerns for writing, I have a technical concern about the estimating $\delta$13C for CO2 emitted from bulk soils (i.e.$\delta$13C RS in the manuscript). The authors estimated $\delta$13C RS using the keeling plots for soil CO2 profile data at 0, 30, 60, 90, and 120 cm depth; however, the$\delta$13C of soil organic matters, the major source of heterotrophic respiration, often change along with soil depth increase. Then, these vertical changes in$\delta$13C of soil organic matters have significant potentials affecting the$\delta$13C-CO2 profile. This means that the observed relationships between$\delta$13C-CO2 and CO2 concentration profiles might be affected not only by the change in contribution of source CO2 and background CO2, but also by the changes in$\delta$13C of source CO2. Therefore, in my opinion, the authors are needed to provide the reliable justification for their methodology, to quantify the uncertainty for estimated$\delta$13C RS, and/or to apply alternative methodology for estimating$\delta$13C RS.

Response: This is indeed an important point and has now been clarified in the revisions. The Ref is correct in noting that the Keeling plot approach is based on 2-end members mixing (as also explained in the Review one of us co-authored; Pataki et al., 2003), and in many cases this assumption does not hold in soils. However, it seems that the very dry conditions at our study site gave us an opportunity to avoid this caveat. As shown in the figure below, there is essentially no change in 13C of soil organics with depth (SD of the 12 samples=0.12 permil; see SI Fig. 2). This is likely because the dry conditions strongly constrain decomposition and probably also the range of microbial

populations (and help explain the high soil carbon storage in this system as noted in the Discussion). It therefore seems that the soil $CO_2$ samplings we carried out still represent predominantly the mixing of atmospheric $CO_2$ with one integrated soil source signal. We must conclude of course that the variations among the contributions of Rsa, Rh, and Ri do not change significantly with depth and the single set of isotopic signatures in Table 2.

Finally, please consider to include the photographs showing conditions of each chamber site and the schematic diagrams describing three collars locations within a chamber site

Response: We are happy to oblige and agree this could help. We added to the SI diagram and photo (see SI Fig. 3).
* * *
[Figure]

**FIGURE 4** Stoniness percentage in L and D plots along the soil profile (left). Root density distribution in L and D plots along the soil profile. Error bars are included, but since their values are low, they are often obscured by the symbols

**Fig. 1.** Figure 1

[Figure]

$\delta^{13}C_h = -24.32 \pm 0.03\,‰$;
$n=12$

**Fig. 2.** Figure 2

[Figure]

**Fig. 3.** Figure 3

---

## Author Response (AR1)

**Detail response to two Reviewers comments, BG-2019-291**

We thank the reviewers for their constructive comments we provide a point by point response below. Note that in general, considering the comments on confusing text, we significantly reduced the text, removing all non-essential parts and sent the manuscript for professional proof editing (Scribandi.com). This included the following changes:

1. Deleting Table 4 (non-essential long-term regressions)
2. Revising Table 3 for clarity including the ratio between the present study and 2003 (mean of 2001-2006).
3. We added material to SI: Expanded Fig. SI-1 as discussed below. New Fig. SI-5 as discussed below. New Fig. SI-8 on heterotrophic and soil respiration as a function of precipitation (based on literature, and used in the Discussion). We added Table SI-3 on literature compilation of background data (used in the Discussion).

We believe these major revisions together with response detail below greatly improved the focus and clarity of the paper and address all points raised by the reviewers. We hope the paper is now ready for final publication.

**Ref1**

This paper uses soil and canopy fluxes with stable isotopes (13C) and radiocarbon(14C) measurements of a 50-year-old dry pine forest over one year to partition the ecosystem's CO2 flux into gross primary productivity (GPP) and ecosystem respiration(Re) and soil respiration flux into autotrophi (Rsa), heterotrophic (Rh), and inorganic(Ri) components. The measurements and data are valuable. The topics are be of great interest.

**Response:** Thank you, it is the most important point. But as we clarify better in the revisions (see also below) the paper goes beyond "one year of partitioning" as this year of measurements allowed us to combine it with our study 10 years earlier at the same site to provide a longer-term perspective on the changes in such partitioning (and as the saying goes, with the "whole is greater than the sum of its parts"…)

But the writing is very confusing.

**Response:** We are sorry this is the case and made a serious effort to fix this by better streamlining the paper (as also noted below), careful proof-reading and, considering that the authors are not native English speakers, also sending it out to professional editing.

Abstract: The abstract lacks critical information.

**Response:** We are not sure what is the missing critical information. We added in the Abstract key information on the site and the study itself. And additional key results after we carefully compared the Abstract and the Result. The results are presented in 3 figures and 4 tables, and carefully comparing the detail summary of the results in the Abstract, indicates that all the main points, including our observed values of GPP, Re, NPP, CUE, the partitioning (the main effort in
this study) of respiration Rs to Rsa to Rh, Ri, the seasonal changes, the main long-term
changes when combined with our earlier study, and the main occlusions.
Introduction: The introduction should be rephrased. There are too much pieces of information on
general knowledge. The Introduction needs a better flow. The scientific significance should be
addressed more. A description that explain why the paper is needed following the previous
studies (i.e. the 2001-2006 study) would be very informative for the readers. Furthermore, I
suggest the authors cite more relevant papers on the Mediterranean climate zones and add one
or two hypothesis.
**Response:** Done. We shortened the introduction by removing two non-critical paragraphs,
which simplify it. We also improved the Motivation at the end of the Intro, and we provide clearer
working hypotheses, as requested. We checked the literature and added missing references on
Mediterranean studies.
Site description: provide more information about the vegetation (e.g. root depth). Flux and
meteorological measurements: How did the gaps in NEE and GPP are filled? How many
missing data points are there due to instrument failure and quality control?
**Response:** Done. We added the requested information, including detail on root depth and on
the vegetation at the site (overstory and understory, indicating the main species). Specifically,
technical details of "gap filling" indicated and a paper with more detail is cited. We also added
information on root depth and note that more detail is provided in our paper published earlier
this year (Preisler et al. 2019, Functional Ecology; now cited):

[Figure]

**FIGURE 4**  Stoniness percentage in
L and D plots along the soil profile (left).
Root density distribution in L and D
plots along the soil profile. Error bars are
included, but since their values are low,
they are often obscured by the symbols

Soil  CO2 flux: I'm really confused. How many data did the author used in the paper? Just one
year? Using just one-year measurements can not identify the long-term temporal changes in the
soil–atmosphere CO2 fluxes in this environment.

**Response:** As noted above, we now clarify this issue straight-out in the Intro and again in the
Methods and in the Discussion. Briefly, note that all the figures and tables indicate that the new
data were obtained during **2015-2016** (one full year). But both the figures/tables and Discussion
show that important virtue of the paper is in combining these new data with our earlier study at
the same site (**2001-2006**) looking at the same parameters and providing the opportunity to
obtain a long-term perspective of the change in the flux component (i.e. in the partitioning) over
a time window of about 10 years. Many of the studies reported in this journal aim at assessing
change, especially in response to global change. While some parameters are monitored
continuously (our flux tower operated continuously for 20 years), other measurements required
for the Partitioning of soil fluxes cannot, practically, be made continuously and here we combine
the continuous measurement with the periodical campaign to assess sufficiently long-term
changes.
Discussion: The hypothesis should also be into Introduction. The present study used only one-
year data, I suggest the authors using a tempered tone in the discussion part. The Discussion
has the same problem with Introduction. There are many pieces of interesting information. But
the discussion should be centered on several key aspects of your results. The Discussion
should echoes the Introduction. I suggest rephrase the Discussion and make a better flow in the
Discussion.
**Response:** Done. The working Hypotheses are included in both the Introduction and
Discussion as noted above, and we streamlined the Discussion deleting some non-critical parts
and to improve the correspondence with the Introduction. As also noted above, we better focus
on the link of the new data of this study with our older study at the same site over 10 years ago
to assess changes to address the issue of changes in the ecosystem and particularly in the soil
carbon flux component over time (we also added in the SI the long-term records of temperature
and precipitations).
**Ref2**
This manuscript describes the study partitioning of canopy and soil $CO_2$ fluxes in a pine forest
at the dry timberline using the measurements of isotopic signatures ($\delta13C$ and $\Delta14C$) of $CO_2$
emitted from bulk soils, fine roots, root-free soils, and carbonate fractions. The measurement
and data are interesting. Then, scientific insights, which can be gained from this study, would
significantly contribute for improving our under-standing the response of dry environment
ecosystems to climate change.
**Response:** Thank you. It is important to see that this study is recognized as adding to
understanding ecosystem response to change.
The writing, however, should be improved more and more as pointed out by Referee #1. Then,
please refine every sentence in the manuscript more carefully, because there are substantial typos (e.g. "a pine forests" in the title, "Soil respiration from the atmosphere" in Line 29-30,
"Reflux" in Line 369, and so on).

**Response:** We recognize that our failure to submit an appropriately proofed manuscript made
some significant damage, although there were no errors in the science. This was the result of
some unfortunate confusion in combining the different versions of the paper proof-read by
different coauthors. It has been fixed and considering that the authors are not native English
speakers the paper has been sent out for professional editing.

In addition to these concerns for writing, I have a technical concern about the estimating δ13C
for CO2 emitted from bulk soils (i.e.δ13C RS in the manuscript). The authors estimated δ13C
RS using the keeling plots for soil CO2 profile data at 0, 30, 60, 90, and 120 cm depth; however,
theδ13C of soil organic matters, the major source of heterotrophic respiration, often change
along with soil depth increase. Then, these vertical changes inδ13C of soil organic matters have
significant potentials affecting theδ13C-CO2 profile. This means that the observed relationships
betweenδ13C-CO2 and CO2 concentration profiles might be affected not only by the change in
contribution of source CO2 and background CO2, but also by the changes inδ13C of source
CO2. Therefore, in my opinion, the authors are needed to provide the reliable justification for
their methodology, to quantify the uncertainty for estimatedδ13C RS, and/or to apply alternative
methodology for estimatingδ13C RS.

**Response:** This is indeed an important point and has now been clarified in the revisions. The
Ref is correct in noting that the Keeling plot approach is based on 2-end members mixing (as
also explained in the Review one of us co-authored; Pataki et al., 2003), and in many cases this
assumption does not hold in soils. However, it seems that the very dry conditions at our study
site gave us an opportunity to avoid this caveat.  As shown in the figure below, there is
essentially no change in 13C of soil organics with depth (SD of the 12 samples=0.12 permil).
This is likely because the dry conditions strongly constrain decomposition and probably also the
range of microbial populations (and help explain the high soil carbon storage in this system as
noted in the Discussion). It therefore seems that the soil CO2 samplings we carried out still
represent predominantly the mixing of atmospheric CO2 with one integrated soil source signal.
We must conclude of course that the variations among the contributions of Rsa, Rh, and Ri do
not change significantly with depth and the single set of isotopic signatures in Table 2.

[Figure]

Finally, please consider to include the photographs showing conditions of each chamber site
and the schematic diagrams describing three collars locations within a chamber site
**Response:** We are happy to oblige and agree this could help. We added to the SI diagram and
photo.

[revised manuscript text omitted]

**(a)** Med.Sea, Yatir Forest, Negev Desert, Judea Desert, Dead Sea
*(adapted from Volcani et al., 2005)*

**(b)** Yatir Forest, Tow
*(adapted from Google maps)*

**Table SI-3 |** Annual heterotrophic respiration (Rh), autotrophic respiration (Ra), soil respiration (Rs), ecosystem
respiration (Re), gross primary productivity (GPP), net primary productivity (NPP), net ecosystem production (NEP; from
eddy covariance), respectively. The relative contribution of Rh, and Ra and Rs to Re, and the ecosystem and soil carbon
sequestration efficiency as CSE (NEP/GPP) and SCSE ($\Delta C_{soil}$/GPP), where $\Delta C_{soil}$ is the annual change of soil carbon.

| Study | Rh | Ra | Rs | Re | GPP | NPP | NEP | Rh/Re | Ra/Re | Rs/Re | CSE | SCSE |
|---|---|---|---|---|---|---|---|---|---|---|---|---|
| | | | | [g m$^{-2}$ y$^{-1}$] | | | | | [%] | | [%] | |
| **Semi-arid[1]** | 115 | 312 | 295 | 488 | 655 | 282 | 167 | 23 | 64 | 60 | 25 | 8.7 |
| **Europe, mean** | 368 | 589 | 657 | 957 | 1107 | 518 | 150 | 38 | 62 | 69 | 14 | 1.8 |
| **Europe[EN], mean** | 461 | 657 | 381 | 1117 | 1475 | 818 | 358 | 41 | 59 | 74 | 24 | 1.4 |
| **Boreal, global** | 301 | 561 | 411 | 862 | 982 | 381 | 116 | 35 | 65 | 48 | 12 | 0.7 |
| **Temperate, global** | 420 | 730 | 773 | 1150 | 1461 | 669 | 306 | 37 | 64 | 67 | 21 | 1.4 |
| **Tropical, global** | 877 | 2184 | 1412 | 3061 | 3351 | 864 | 403 | 29 | 71 | 46 | 12 | 0.3 |

[1] This study from November 2015 to October 2016. [EN] Evergreen needleleaf forests. References for all other vegetation
types appear in the SI.

[Figure]

**Figure** SI-1 | (a) Landsat-TM image of Central Israel. (b) Map of the experimental set-up at the *Pinus halepensis*

Yatir forest with white rectangle for soil respiration (Rs) measurements and white dot is the  eddy
covariance tower (NEE), c, d, and e) Photographs showing conditions of locations, site, and the schematic diagrams
describing experiment design.

[Figure]

**Figure SI-2 |** The linear regression line used to estimate the $\Delta^{14}C$ of Rs. The line (dotted) was produced by the
correlation between the average of the measured $\delta^{13}C$ values of Rsa, Rsh, and the $\delta^{13}C$ Ri (all from incubation
measurements), and the $\Delta^{14}C$ values estimated based on measured $\Delta^{14}C$ at our site (Carmi et al. 2013) adjusted to the
present study period and the mean accepted ages of autotrophic and heterotrophic soil organic material (Graven et al.,
2012; Levin et al., 2010; Taylor et al., 2015).

[Figure]

**Figure SI-3 |** Monthly averages of $\delta^{13}C$ (‰) from the soil $CO_2$ profile (at 0, 30, 60, 90, and 120 cm soil depth)
during some campaigns in 2016 to determine the seasonal variations in the relative contribution of soil autotrophic (Rsa),
heterotrophic (Rh), and abiotic (Ri) components to Rs.

[Figure]

**Figure S4SI-4 |** Keeling plot for soil $CO_2$ profile (at 0, 30, 60, 90, and 120 cm soil depth) during some campaigns in
2016 to determine the seasonal variations in the relative contribution of soil autotrophic (Rsa), heterotrophic (Rh), and
abiotic (Ri) components to Rs.

[Figure]

**Figure SI-5** | $\delta^{13}C$ of soil organics profile with depth (at 0-5, 5-10, 10-20, and 20-50 cm soil depth) from three sites during some campaigns in 2016 to determine the relative contribution of soil heterotrophic (Rh) to Rs (STDEV of the 12 samples = 0.12‰).

[Figure]

**Figure S5SI-6 |** a) half-hour values for soil temperature 5 cm (Ts) and soil water content 10 cm ($SWC_{0-10cm}$), b) half-hour values for the air temperature at 20 cm (Ta) and relative humidity at 20 cm (RH), c) daily average of incoming photosynthetic activity radiation above canopy (PAR) and vapour pressure deficit (VPD), half-hour values for the following $CO_2$ fluxes d) up-scaled Rs, e) ecosystem respiration (Re), f) gross primary production (GPP), and g) net ecosystem exchange (NEE). Black lines are a running average lines for a widows of 2 days.

[Figure]

[Figure]

**Figure S6SI-7 |** Typical diurnal cycle of the meteorological parameters during the wet period (Nov.-Apr.; upper panels) and for the dry period (May-Oct.; lower panels); each set includes six months of half-hour measurements. a and e) incoming photosynthetic activity radiation above canopy (PAR) and vapour pressure deficit (VPD), b and f) wind speed (WS) and covariation of friction velocity (U*), c and g) soil temperature at 5 cm (Ts) and air temperature at 20 cm (Ta), and d and h) relative humidity (RH) and soil water content at the top 10 cm (SWC0-10cm). Shaded areas indicate ±se.

[Figure]

**Figure SI-8 |** An asymptotic function based on the Michaelis-Menten equation (Kool et al., 2007) was fit to a) Rs or b) Rh vs. MAP from the European Evergreen needleleaf data as follows: Rs = MAP/(0.6669 + MAP/2240), $R^2$ = 0.65, $p$ < 0.01, n=13 (Flechard et al., 2019a).

**Supplementary References (used for data in Table SI-3)**

1.  Etzold, S., Ruehr, N. K., Zweifel, R., Dobbertin, M., Zingg, A., Pluess, P., et al. (2011). The Carbon Balance of Two Contrasting Mountain Forest Ecosystems in Switzerland: Similar Annual Trends, but Seasonal Differences. Ecosystems, 14(8), 1289-1309.
2.  Flechard, C. R., Ibrom, A., Skiba, U. M., de Vries, W., van Oijen, M., Cameron, D. R., et al. (2019). Carbon / nitrogen interactions in European forests and semi-natural vegetation. Part I: Fluxes and budgets of carbon, nitrogen and greenhouse gases from ecosystem monitoring and modelling, Biogeosciences Discuss., https://doi.org/10.5194/bg-2019-333, in review, 2019.
3.  Flechard, C. R., van Oijen, M., Cameron, D. R., de Vries, W., Ibrom, A., Buchmann, N., Dise, N. B., et al. (2019). Carbon / nitrogen interactions in European forests and semi-natural vegetation. Part II: Untangling climatic, edaphic, management and nitrogen deposition effects on carbon sequestration potentials, Biogeosciences Discuss., https://doi.org/10.5194/bg-2019-335, in review, 2019.
4.  Hursh, A., Ballantyne, A., Cooper, L., Maneta, M., Kimball, J., & Watts, J. (2017). The sensitivity of soil respiration to soil temperature, moisture, and carbon supply at the global scale. Global Change Biology, 23(5), 2090-2103. doi:10.1111/gcb.13489
5.  Schulze, E. D., Luyssaert, S., Ciais, P., Freibauer, A., Janssens, I. A., Soussana, J. F., et al. (2009). Importance of methane and nitrous oxide for Europe's terrestrial greenhouse-gas balance. Nature Geoscience, 2(12), 842-850.
6.  Avitabile, V., & Camia, A. (2018). An assessment of forest biomass maps in Europe using harmonized national statistics and inventory plots. Forest Ecology and Management, 409, 489-498.
7.  De Vos, B., Cools, N., Ilvesniemi, H., Vesterdal, L., Vanguelova, E., & Camicelli, S. (2015). Benchmark values for forest soil carbon stocks in Europe: Results from a large scale forest soil survey. Geoderma, 251, 33-46.
8.  Luyssaert, S., Inglima, I., Jung, M., Richardson, A. D., Reichstein, M., Papale, D., et al. (2007). CO2 balance of boreal, temperate, and tropical forests derived from a global database. Global Change Biology, 13(12), 2509-2537.
Pan, Y. D., Birdsey, R. A., Fang, J. Y., Houghton, R., Kauppi, P. E., Kurz, W. A.,

[Figure]

**Figure S7 |** Representative diurnal cycles of the ambient $H_2O$ and $CO_2$ concentrations at ground level during a) the wet (Nov-Apr) and b) the dry (May-Oct) periods, each set includes six months of half-hour measurements. These concentrations were measured with the system that determined Rs; shaded areas indicate ±se.

14.9.    et al. (2011). A Large and Persistent Carbon Sink in the World's Forests. Science, 333(6045), 988-993.

---

## Author Response (AR2)

**Detail response to the Reviewer comments, BG-2019-291**

We thank the reviewer for the comments and for identifying some remaining issues. Below we provide a point by point response essentially adopting all comments and suggestions. We hope make the paper ready for publication.

The quality of manuscript is significantly improved by authors, and is now close to the publication in the journal.

**Response:** Thank you.

Before the accept, I feel there is a little bit more of needs of revision and/or re-consideration for some specific points in the manuscript. Some of them are very confusing for readers to understand the manuscript easily.

Those specific points are as follow:
L13: "measured" should be "were"

**Response:** Corrected.

L121: "An instrumented eddy covariance tower" should be "An instrumented eddy covariance (EC) tower"

**Response:** Corrected.

L224: In order to secure reliability of the Keeling plot approach in the present study site, add a sentence, which is similar to the sentence in L166-L167 of Author's response (i.e. "We must conclude of course that...."), after the sentence ended with "....to avoid this caveat".

**Response:** We added the sentence as suggested.

L256: "The analytical precision was 0.1%" should be "The analytical precision was 0.1‰"?

**Response:** Corrected.

L269: "The precision was 0.1%" should be "The precision was 0.1‰"?

**Response:** Corrected.

L271-L286: Because "RI" in Eq.12 is very confusing with "RI" in Table 3 (RI means
foliage respiration here), "RI" in Eq.12 should be "ALP" or something else which makes
it to be easier to imagine the aboveground litter production.
**Response:** As suggested, we replaced the RI with $R_{alp}$ for the aboveground litter
production.
L305: "r = 0.62" should be "r = -0.62".
**Response:** Corrected.
L306: "r = 0.45" should be "r = -0.45".
**Response:** Corrected.
L337: The sentence "daily Rs values could reach 6.1 µmol m-2 s-1" is true? There is no
Rs data over the value of 5 µmol m-2 s-1 in Figure S1-6 nor Figure 2.
**Response:** Checked and this value is correct but refers to the under trees (UT)
microsite while Fig. 2 shows integrated mean values. To clarify the sentence we added
"**i.e. in the UT microsite; data not shown**".
L340: Delete "(Fig. S1-6)".
**Response:** Corrected. It refers to SI Fig. 6.
L349: Add "(no data in any of figures nor tables)" after "(r = 0.2 and 348 0.1,
respectively; p < 0.01)".
**Response:** We added the sentence.
L351: Add "(no data in any of figures nor tables)" after "(CV~40%)".
**Response:** Added as suggested: **"(correlations and CV values were not included in
figures and tables)**".
L359: "Refluxes" should be "Re fluxes".
**Response:** Corrected.

L373: "being highest" should be "being the highest".
**Response:** Corrected.
L377: "Repartitioning" should be "Re partitioning".
**Response:** Corrected.
L382: Add "(Fig. 2)" after "limited water loss".
**Response:** We added the (Fig. 2).
L409: What does "Despite relatively high rates of respiration fluxes" mean? According
to Table 3, the magnitude of Re is only 75% of GPP. Add clear description for
"relatively high rates of respiration fluxes".
**Response:** Thank you. The sentence was revised for clarity.
**"These rates of respiration fluxes translated at the ecosystem scale to Re/GPP of**
**~75%, lower than observed in other ecosystems (SI Table 3) and leading, in turn,**
**to high ecosystem CUE of 0.43"**
L417-L418: "presented n Table 3" should be "presented in Table 3".
**Response:** Corrected.
L418-L419: "the belowground allocation" should be "the TBCA".
**Response:** Corrected.
L419-L423: This sentence is very confusing. Does this sentence mean that "While there
is little change in the ratio of the respiration components to Re or to GPP, the shift from
the autotrophic components (i.e. Rsa, Rl, and Rw) to the heterotrophic component (i.e.
Rh) has occurred as indicated by increasing ratio of Rh to Rsa + Rl + Rw from 0.27 in
2003 to 0.37 in 2015." ?
**Response:** We revised this sentence for clarity.
L467: "the remarkable increase" may be prefer than "the marked increase".

144 **Response:** Wording changed.

147 L468-L472: This sentence would be better to be changed as "Here, because comparing

148 the non-continuous data from the present (2016) and earlier (2001–2006) studies is

149 sensitive to the large interannual variations in the ecosystem activities and fluxes

150 (Qubaja et al., in press), we focused on the more robust changes in the ratio of the

151 respiration components to the overall fluxes (Re) (Table 3)".

153 **Response:** Wording changed as suggested.

156 Figure 3: Caption should be "a) The seasonal variations in the relative contribution of

157 soil autotrophic (Rsa), heterotrophic (Rh), and abiotic (Ri) components to Rs, and b)

158 seasonal variations in the relative contribution of soil autotrophic (Rsa), heterotrophic

159 (Rh), abiotic (Ri), and foliage and stem respiration (Rf is obtained from the Re-Rs)

160 components to ecosystem respiration (Re) during eight campaigns (Jan–Sep) in 2016.

161 The contributions were determined with linear mixing models using isotope signature

162 analysis ($\delta 13C$ and $\Delta 14C$) of soil $CO_2$ profile from 0 to 120 cm soil depth. These

163 results confirmed earlier estimates of Grünzweig et al. (2009) and Maseyk et al.

164 (2008a)".

166 **Response:** The caption was re-arranged as suggested.

169 Figures 2, 3, SI-6: Please consider to add vertical lines representing the periods of

170 spring, summer, autumn, and winters. In the manuscript, seasonal characteristic of

171 fluxes is one of the mainly focused topics; however, it is difficult to identify those from

172 figures without the visually representing period of individual season.

174 **Response:** We added the vertical lines as suggested and indicate it in the captions

177 Finally, please run careful refining of every sentences in the manuscript again, in order

178 to avoid unnecessary confusion.

180 **Response:** Done

[revised manuscript text omitted]